# Expanding the Diversity of *Myoviridae* Phages Infecting *Lactobacillus plantarum*—A Novel Lineage of *Lactobacillus* Phages Comprising Five New Members

**DOI:** 10.3390/v11070611

**Published:** 2019-07-04

**Authors:** Ifigeneia Kyrkou, Alexander Byth Carstens, Lea Ellegaard-Jensen, Witold Kot, Athanasios Zervas, Amaru Miranda Djurhuus, Horst Neve, Martin Hansen, Lars Hestbjerg Hansen

**Affiliations:** 1Department of Environmental Science, Aarhus University, Frederiksborgvej, 399, 4000 Roskilde, Denmark; 2Department of Plant and Environmental Sciences, Thorvaldsensvej 40, 1871 Frederiksberg, Denmark; 3Department of Microbiology and Biotechnology, Max Rubner-Institut, Hermann-Weigmann-Straße 1, 24103 Kiel, Germany

**Keywords:** *Lactobacillus plantarum*, phage, new genus, annotation, comparative genomics, phylogenetics, isolation, diversity

## Abstract

*Lactobacillus plantarum* is a bacterium with probiotic properties and promising applications in the food industry and agriculture. So far, bacteriophages of this bacterium have been moderately addressed. We examined the diversity of five new *L. plantarum* phages via whole genome shotgun sequencing and in silico protein predictions. Moreover, we looked into their phylogeny and their potential genomic similarities to other complete phage genome records through extensive nucleotide and protein comparisons. These analyses revealed a high degree of similarity among the five phages, which extended to the vast majority of predicted virion-associated proteins. Based on these, we selected one of the phages as a representative and performed transmission electron microscopy and structural protein sequencing tests. Overall, the results suggested that the five phages belong to the family *Myoviridae*, they have a long genome of 137,973–141,344 bp, a G/C content of 36.3–36.6% that is quite distinct from their host’s, and surprisingly, 7 to 15 tRNAs. Only an average 41/174 of their predicted genes were assigned a function. The comparative analyses unraveled considerable genetic diversity for the five *L. plantarum* phages in this study. Hence, the new genus “Semelevirus” was proposed, comprising exclusively of the five phages. This novel lineage of *Lactobacillus* phages provides further insight into the genetic heterogeneity of phages infecting *Lactobacillus* sp. The five new *Lactobacillus* phages have potential value for the development of more robust starters through, for example, the selection of mutants insensitive to phage infections. The five phages could also form part of phage cocktails, which producers would apply in different stages of *L. plantarum* fermentations in order to create a range of organoleptic outputs.

## 1. Introduction

*Lactobacillus plantarum* is a gram-positive, non-sporeforming, lactic acid bacterium with probiotic qualities. It is commonly encountered in a variety of environments ranging from dairy products, meat, grape must and vegetable fermentations to the human mouth, gastrointestinal tract and stool, as well as sewage and cow dung [1]. Lately, the species has received attention as a promising biocontrol agent in agriculture due to its favourable stimulating effects on crop growth and yield, and its activity against phytopathogenic microbes [2]. From a winemaking perspective, *L. plantarum* has been proposed and utilised as an alternative malolactic fermentation (MLF) starter [3,4,5]. This is thanks to its wide variety of enzymes linked with aroma compounds, and its production of bacteriocins against spoilage bacteria [3]. Other characteristics of *L. plantarum* that render it an attractive MLF starter are its ability to endure conditions of low pH, high ethanol, sulfur dioxide, low temperatures, fatty acids, tannins and lysozyme [6]. Lastly, like other lactic acid bacteria (LAB), *L. plantarum* can both be a traditional preservative of fermented food products [1] and a food spoiler [7].

While fermentation starter strains of *L. plantarum* and other LAB are of great industrial value, their use is not challenge-free. Bacteriophage (phage) attacks often hamper their activity and by extension cause economic losses because of fermentation failure or poor product quality [8]. Owing to the acute threat they pose to industrially-relevant starters, *Lactococcus* and *Streptococcus* phages have been long and extensively studied [9,10,11]. On the other hand, the knowledge gaps on the biology of *Lactobacillus* phages and their genetic heterogeneity have negatively impacted taxonomic efforts [12,13]. More than 200 published *Lactobacillus* phages have never been sequenced [13], and of the 56 complete genome sequences just eight are formally classified (NCBI search). Nevertheless, it is clear that research on *Lactobacillus* phages is being revitalised. In their review in 2017, Murphy et al. [8] documented 27 complete genomes of *Lactobacillus* phages; a year on, this number has been doubled.

As a subset of *Lactobacillus* phages, *L. plantarum* phages have also been affected by the paucity of information and the moderate earlier interest in their host. The first reference on virulent phages infecting *L. plantarum* dates back to 1969 [14]. Since then, several virulent and temperate phages of *L. plantarum* have been discovered. A few of these phages were reported to have a high level of strain specificity [15]. Transmission electron microscopy (TEM) images of *L. plantarum* phages have principally placed them in the family *Siphoviridae*. The phages fri [16], phiPY1 and phiPY2 [17], phi22-D10 [18], and LP65 [19] are all virulent and are currently the sole representatives of the family *Myoviridae*, and a single account of a *Podoviridae* phage [20] is probably inaccurate, according to Villion and Moineau [13]. Confirmed sources of phages against *L. plantarum* are sewage, feces, silage, kimchi, sauerkraut, fermented coffee, chicha, whey, pear, cucumber, plant materials, salami and meat [13,19,21]. Noteworthy is the latest publication on the temperate phage PM411, as this phage represents the first to be associated with an agriculturally-relevant strain of *L. plantarum* [21]. In this study, we describe new *L. plantarum* phages isolated from organic household waste, and we provide further insight into the diversity of phages of *Lactobacillus*. On a global scale, our aim is to broaden current knowledge on the viral tree of life through a novel group of phages that infect *Lactobacillus*, a genus of probiotic bacteria with emerging roles in agriculture, as well as in the dairy and wine industries.

## 2. Materials and Methods

### 2.1. Bacterial Strains and Culture Media

Two bacterial strains of *L. plantarum*, strains L1 and MW-1, were used as indicator strains for phage isolation, propagation and characterisation. Both L1 and MW-1 were obtained from private collections and had initially been isolated from a wine fermentation sample and grapes, respectively. Bacterial cultures and phage propagations were performed in De Man, Rogosa and Sharpe (MRS) broth and agar (Difco Laboratories, Detroit, MI, USA). All cultures and propagations were grown overnight at 37 °C without shaking to ensure minimum aeration, unless otherwise stated.

### 2.2. Environmental Sampling, Isolation, Purification, and Enrichment of Phages

Organic household waste samples (mainly consisting of food-related waste) were collected from two different organic household waste treatment plants (hereafter named treatment plants A and B) in Denmark in February and May 2017 and split into two subsamples. The first part of the samples was centrifuged (5000× *g*, 15 min, 25 °C) and supernatants were passed through 0.45-μm pore size PVDF syringe filters (Merck Millipore, Darmstadt, Germany). The filtrates were stored at 4 °C until use. The other part was centrifuged (5000× *g*, 15 min, 25 °C) and phages were precipitated with polyethylene glycol (PEG). Specifically, PEG 8000 (10% *w*/*v*) was dissolved in 250 mL of supernatant, stored for 1 h at 4 °C and then centrifuged (10000× *g*, 10 min, 4 °C). Formed pellets were dried for 15 min, carefully resuspended in 5 mL of SM buffer (0.05 M TRIS, 0.1 M NaCl, 0.008 M MgSO_4_, pH 7.5) and stored overnight at 4 °C. Following that, they were treated as explained for the first part of samples, which resulted in concentrated filtrates. The presence of bacteriophages in both plain and concentrated filtrates was assessed using a double agar overlay assay [22]. Briefly, 0.1 mL of one indicator strain (10^6^–10^7^ colony-forming units (CFU)/mL) and 0.1 mL of one filtrate at a time were added in MRSφ, i.e., MRS broth supplemented with 0.4% *w*/*v* agarose and 10 mM CaCl_2_. Mixtures were poured on top of MRS agar and incubated overnight at 25 °C. To select for phages specific to *L. plantarum*, 16 mL of the plain filtrates were blended with 20 mL of 2× MRSφ and supplemented with 4 mL of an overnight culture of either strain L1 or strain MW-1. The mixtures were incubated overnight, and their filtrates tested for phages using the aforementioned overlay assay.

For the purification of phages, all plates deriving from the overlay assay were examined and single plaques were picked. Each plaque was then transferred into 0.7 mL of SM buffer, filtered through 0.45-μm pore size PVDF spin filters (Ciro, FL, USA) and propagated on the same strain. Occasionally, 100 mM glycine were also added to promote plaque formation and visibility [23]. This procedure was repeated twice to ensure purity and phage stocks were stored at 4 °C. Generally, high titers were produced by infecting 0.1 mL of the indicator strains (10^6^–10^7^ CFU/mL) with 0.5 mL of each phage isolate (10^5^–10^6^ plaque-forming units (PFUs)/mL) in 10 mL of MRSφ. In some cases though, phage lysates of high titer were produced as elaborated elsewhere [24] and kept at 4 °C until needed. The filtered phage lysates were further purified by a caesium chloride gradient according to the protocol of Sambrook [25].

### 2.3. Phage DNA Extraction, Library Preparation, and Sequencing

For the DNA extraction, 0.3 mL of filtered phage lysates ranging from 10^8^–10^10^ PFUs/mL were used and the extraction was performed following a standard phenol/chloroform protocol [26]. DNA pellets were resuspended in 2 µL of TE buffer (10 mM Tris–HCl, 0.1 mM EDTA, pH 7.5). Libraries were built with the Nextera® XT DNA kit (Illumina Inc., San Diego, CA, USA) and later sequenced on an Illumina MiSeq as a part of the flowcell using the MiSeq v2, 2 × 250 cycle chemistry. The library normalisation, pooling, and sequencing were carried out as described elsewhere [27].

### 2.4. Assemblies and Annotations

Illumina reads from the DNA sequencing were cleaned using VecScreen (NCBI) and Cutadapt (v. 1.8.3) [28] and assembled with SPAdes (v. 3.5.0) [29] as detailed in [30]. SPAdes assemblies were cross-verified by Unicycler (v. 0.4.3) [31]. Furthermore, assemblies were compared against those generated by CLC Genomic Workbench (v. 9.5.3; CLC bio, Aarhus, Denmark), by first applying the “merge overlapping pairs” tool for overlapping reads and subsequently the “trim sequences” and the “de novo assembly” tools. Phage genomes were auto-annotated by the RAST annotation server v. 2.0 [32] using the RASTtk annotation scheme and GeneMark [33] as a gene caller. Protein functions were ascribed only after manually corroborating RASTtk predictions with BLASTp [34] and HHpred [35], and occasionally with Pfam [36], TMHMM [37] and DELTA-BLAST [38]. The tool tRNAscan-SE (v. 2.0) [39] was used to search for existing tRNA genes. All phage genomes were scanned against ISFinder to identify insertion sequences using BLASTn and the default pipeline of ISFinder [40].

### 2.5. Transmission Electron Microscopy

The caesium chloride-treated phage lysate was adsorbed to freshly prepared ultra-thin carbon film and treated with 1% (*v*/*v*) EM-grade glutaraldehyde (20 min) for fixation. Subsequently, negative staining was done with 2% (*w*/*v*) uranyl acetate. The specimen was picked up with 400-mesh copper grids (Plano, Wetzlar, Germany) and analysed using a Tecnai 10 transmission electron microscope (Thermo Fisher, Eindhoven, the Netherlands) at an acceleration voltage of 80 kV. Micrographs were taken with a MegaView G2 CCD-camera (EMSIS, Muenster, Germany).

### 2.6. Identification of Structural Proteins

To identify the proteins, we followed a procedure described elsewhere [41] with minor modifications. In short, 100 µL of the caesium chloride-treated phage lysate were transferred to Amicon Ultra filter unit (MWCO 30k Da) and centrifuged at 14,000× *g* for 20 min and further desalted four times with 450 µL water. The filtrate containing the phage particles (10 µL) was denaturised in 25 µL buffer consisting of 6 M urea, 5 mM dithiothreitol and 50 mM Tris-HCl (pH 8). The phage particles were destabilised by five successive freeze-thawing cycles followed by an incubation at 60 °C for an hour to reduce the phage proteins. The proteins were alkylated by adding 25 µL 100 mM iodoacetamide and 150 µL 50 mM ammonium bicarbonate, then incubated for 45 min at room temperature. Phage proteins were digested with 0.8 µg trypsin dissolved in 40 µL 50 mM ammonia bicarbonate and incubated for a day at 37 °C. The protein digest was diluted 1:1 with 0.05% trifluoroacetic acid. The phage peptides were analysed with an Ultimate 3000 RSLCnano UHPLC system hyphenated with a Q Exactive HF mass spectrometer (ThermoFisher Scientific, Denmark). An amount of 21.4 µL of sample was loaded on a pre-concentration trap (a C_18_ 100 µm × 5 mm cartridge) and eluted onto an analytical column (75 µm × 250 mm, 2 µm C_18_) with a chromatographic triple-phasic 53 min gradient ranging from 1% to 64% mobile phase B (98% acetonitrile and 0.1% formic acid) at 300 nL per minute. The total analysis time was 65 min, and mobile phase A consisted of 2% acetonitrile and 0.1% formic acid. The high-resolution mass spectrometer was operated with positive electrospray ionisation in data-dependent mode by automatically switching between MS and MS/MS fragmentation. Based on a survey MS scan in the Orbitrap, operated at a mass resolution of 120,000 at m/z 200 with a target of 3e6 ions and a maximum injection time at 50 ms, the twelve most intense peptide ions were selected for MS/MS fragmentation in subsequent scans. The selected ions were isolated (in a m/z 1.4 window) and higher-energy collision dissociation was done at a normalised collision energy (28), and fragments recorded in centroid mode at a resolution of 60,000 (m/z 200) with a 250 ms max filling time and target of 1e5 ions. The generated high-resolution data was analysed in a Proteome Discoverer 2.2 (ThermoFisher Scientific) and searched against predicted phage proteins by the Sequest HT algorithm in an iterative processing pipeline. The search criteria were: enzyme, trypsin (full); dynamic modifications, methionine oxidation and acetyl (N-terminus); precursor mass tolerance, 5 ppm; and, fragment mass tolerance, 20 mDa. The processed data were filtered in a Proteome Discoverer consensus workflow with the Peptide Validator algorithm (*q*-value < 0.01) to ensure the peptide-spectrum match had a false discovery rate under 1%.

### 2.7. Comparative Genomics and Phylogenetics to Distant Relatives

In accordance with the International Committee on Taxonomy of Viruses (ICTV) [42], each phage sequence was initially compared to the viral nucleotide collection database (taxid:10239; nt/nr) using the BLASTn algorithm. Phage records that showed some degree of similarity with the new phage isolates of this study were selected and included in an all-against-all, quantitative DNA comparison with Gegenees (v. 2.1) [43]. The Gegenees analysis generally relies on the division of a full genome (query) into fragments and the search for BLAST “seeds” of each fragment against another genome (reference) [43]. The final phylogenomic distance of a query from a reference is the average value of all fragments’ BLASTn scores but expressed as a percentage of the score each fragment would yield towards itself (at 100% identity). In this study, the Gegenees analysis was performed with a customised setting of 50/25, fragment size/sliding step size, and an “accurate” threshold of 0%. In addition, two phylogenetic trees, one for the major capsid protein and one for the large subunit terminase, were constructed using the default pipeline of “One Click mode” (http://phylogeny.Lirmm.fr/) [44]. The tree leaves comprised the phages of this study and selected phage records from the Gegenees comparisons, which had earlier yielded average nucleotide similarity of at least 0.05 or higher with Gegenees. Homology between and within proteomes was assessed via a BLASTp comparison of the new phages and their closest phage relative using the CMG-biotools system [45]. In this type of analysis, a pair of amino acid sequences is considered homologous (paralogous or orthologous) if the length of the alignment is at least 50% of the longest sequence and the identity of the alignment is at least 50%. Paralogue hits were determined using MAFFT [46] (v. 7.388) with the following arguments: algorithm FFT-NS-i x1000; scoring matrix 200PAM/k = 2; gap open penalty 1.53; offset value 0.123, and then BLASTp. Finally, the genomes of the new phages and the two most related phage genomes were examined for common synteny with Easyfig by tandemly comparing phage pairs with BLASTn [47].

### 2.8. Phage Genomic Data Availability

Assembled and annotated genomes of Semele, Bacchae, Iacchus, Dionysus and Bromius have been uploaded to GenBank under accession numbers MG765279, MG765277, MH809529, MH809530 and MH809531, respectively.

## 3. Results and Discussion

### 3.1. Isolation of Phages and Basic Features

Each one of the phages Semele, Bacchae, Iacchus, Dionysus and Bromius was isolated from a different in vitro treatment of the original organic household waste samples (treatment details in Section 2.2.). Phage Bacchae was isolated from the sample of treatment plant A, and Semele, Iacchus, Dionysus and Bromius came from the sample collected from treatment plant B. All five isolates formed clear plaques with a diameter of approximately 0.5 mm on their respective host strains, L1 and MW-1 (see Appendix A for an example of a plate with plaques). In particular, phages Semele and Bacchae were isolated from an infected lawn of strain L1, while phages Iacchus, Dionysus and Bromius were isolated from an infected lawn of strain MW-1.

Transmission electron microscopy imaging analysis was carried out with a representative phage, i.e., phage Dionysus (Figure 1, Table 1). A representative phage was chosen due to the five phages being quite unstable and difficult to propagate at high titers. Nevertheless, Dionysus can sufficiently represent the studied phages because the vast majority of morphogenesis genes of the five phages were proven to be highly homologous (please read Section 3.3. and Appendix A for comparisons with the virions of phages Semele and Bacchae). The only obvious exception was one gene predicted to encode a tail protein in all five phages (AUV60228 of Semele, AYH92260 of Bromius, AUV59956 of Bacchae, AYH92087 of Dionysus and AYH91916 of Iacchus). Dionysus revealed a remarkably high stability, as only a low number of phage particles with contracted tail sheaths were detected (Figure 1b). The phage has a large isometric capsid (diameter: 93 nm) with characteristic (“blackberry-like”) surface decorations. At the distal end of the (contractible) tail (length 222 nm), a complex baseplate structure is visible, and unique flexible appendages (length ca. 22 nm) with terminal globular structures (ca. 9 nm in diameter) are attached under the baseplate. A flexible thin tail fiber (length: 30 nm) terminating with approximately three distinct knob-like structures protrudes under the baseplate. These morphological characteristics classify phage Dionysus into the order *Caudovirales* and the family *Myoviridae*. This is in agreement with existing records of phages infecting *L. plantarum*, all of which fall into the order *Caudovirales*, while five of those belong to the family *Myoviridae* [13].

All five genomes of this study were assembled into a single major contig of size range 137,973–141,344 bp, G/C content range 36.3–36.6%, and with average coverage range 77.1–600.6× (Table 2). The noted genome size is close to the longest observed genome for *L. plantarum* phages (145,162 bp, phage LpeD, accession number MF787246.1).

### 3.2. Analyses of DNA Sequences and Protein Predictions

Annotations of Semele, Bacchae, Iacchus, Dionysus and Bromius genomes assigned functions to 40 of the 177, 38 of the 180, 44 of the 170, 42 of the 172, and 39 of the 172 open reading frames (ORFs), respectively. Of the total number of ORFs, one-third is harboured by the antisense strand. These genomes were also shown to encode clusters of 11, 15, 7, 7, and 7 tRNA genes, accordingly. Additional analyses presented in Section 3.3 suggest that these phage isolates are new species and could be members of a new *Lactobacillus* phage genus. Below, we discuss the different categories of proteins produced by these five phage genomes.

#### 3.2.1. Transcription and Translation Takeover

A module of 7–15 tRNA-encoding genes, often accompanied by between one and three repeat regions, points to a potential mechanism of host transcription and translation takeover in all five *Lactobacillus* phage genomes. These tRNAs are predicted to carry 7–11 different amino acids, with a common core of seven amino acids, with the first two (Gly, Thr) and the final (Arg) maintaining their position across modules (please see Appendix A for details on the order of tRNAs and their frequency). Specifically, Semele’s tRNAs carry the amino acids Gly, Thr, Trp, Met, Phe, Leu, Asn, Ile, Tyr, and Arg; Bacchae’s tRNAs deliver the amino acids Gly, Thr, Trp, Met, Phe, Leu, Gln, Ser, Asn, Ile, and Arg; Iacchus’, Dionysus’ and Bromius’ tRNAs deliver the amino acids Gly, Thr, Leu, Asn, Ile, Phe, and Arg. Interestingly, similar observations were made for the genome of the distantly related phage LP65 of *L. plantarum*, wherein 14 tRNAs were identified. In particular, the equivalent LP65 module has the same core of tRNA genes and shares the same first two and one last amino acids. Additionally, all five phages in this study possess two additional transcription-related elements, an RNA ligase and an RNA polymerase sigma factor. The RNA ligase is homologous to the T4 RNA ligase by the gene *mlA* (Pfam domain search). This enzyme has a double activity in T4. It participates in the repair of tRNAs to reverse the damage inflicted by host defense enzymes and it catalyses the attachment of tail fibers [48]. An RNA polymerase sigma factor has been described in phages before. In T4, the phage-encoded sigma factor serves the binding of the RNA polymerase to promoters different from the ones recognized by the host sigma factor [49]. In other words, the two sigma factors direct the transcription of the phage DNA independently and are complementary to each other. A hypothetical connection between the T4 sigma factor and the phage’s DNA packaging has also been described [50].

#### 3.2.2. DNA Metabolism, Replication, Recombination, and Repair

The five phage genomes seem to have a substantial DNA replication, recombination, and repair system, which is fundamentally shared among them (Appendix A). In confirmation of earlier observations, the genes of this system are arranged in modules [48] and are situated adjacent to DNA metabolism genes [48]. The replisome genes encode a DNA primase, a DNA polymerase, one or two DNA helicases, one or two DNA-binding proteins of unknown function and a recombinase A, not to mention putative recombination or repair-related proteins. Not least, a nucleotide precursor complex suggests that these phages may manipulate the nucleotide pool of their hosts and regulate DNA biosynthesis independently. Such a mechanism would be crucial given that their G/C content (Table 2) differs considerably from their host’s (~44.5%) [48]. Indeed, the predicted phosphatase enzymes phosphatase/phosphodiesterase and deoxynucleoside kinase could allow for the dephosphorylation [16] and phosphorylation [17] of nucleotides, accordingly. Moreover, a gene associated with the nucleoside deoxyribosyltransferase of Lactobacillus phage ATCC 8014-B2 could catalyse the transfer between purines and/or pyrimidines, just as found for many lactobacilli, and thus contribute to nucleotide recycling [51]. The array of genes for a nicotinamide riboside transporter, a nicotinamide-nucleotide adenylyltransferase and ADP-ribose pyrophosphatase may be involved in a potential pyridine nucleotide (NAD+) salvage pathway. In vitro tests to support an analogous observation have recently been performed for a *Vibrio* bacteriophage [52]. A gene producing a nicotinamide mononucleotide transporter, present in all the phages studied here, may be another unit of this pathway. At least 35 phages of different bacterial species have a nicotinamide mononucleotide transporter, including *L. plantarum* phages LpeD and LP65, but its exact functionality has not been verified [53]. The only information available suggests that this transporter can pass nicotinamide mononucleotide molecules across membranes and that it is similar to its bacterial counterpart produced by *L. plantarum* [53]. The latter hints at a recent gene transfer between host and phages [53]. The fact that the five phages and their host share a gene, and the striking difference in G/C content among the five phages and *L. plantarum* may mean that these phages have a host range beyond *L. plantarum*. Hosts other than their primary host have been demonstrated for *L. plantarum* phages before [17,18,19].

#### 3.2.3. Self-Splicing/Selfish Genetic Elements

Self-splicing/selfish genetic elements are well-represented and interspersed throughout the genomes of the five phages (Appendix A). One example is an unknown mobile genetic element protein, conserved in four of the five phages of this study and typically placed next to and upstream of the tRNA module. Another instance is that of transposases. Two transposases are observed in the genomes of Dionysus, Semele and Iacchus, and their neighbouring putative AAA ATPase probably controls their transposition to target DNAs [54]. None of these transposases returned any significant similarity to entries of inserted sequence elements in ISFinder [40]. Since lysogeny is widespread in lactobacilli [55], transposable genetic elements from the chromosome could partially drive the evolution of virulent *Lactobacillus* phages from temperate ones, as reported for Lactobacillus phage φFSW [9]. However, a BLASTn search against the bacterial database did not return any similarity of Dionysus, Semele or Iacchus to prophage records. In the genomes of Semele and Iacchus, a protein of unknown function splits the DNA polymerase gene into two fragments. This protein cannot be a genetic switch since it has the same direction as the DNA polymerase, but it may either be part of a self-splicing group I intron or a mini-intein that could regulate the expression of the DNA polymerase. Group I introns are ribozymes capable of self-splicing from primary transcripts [56]. They interrupt tRNA, rRNA, and protein-coding genes and sometimes contain their own homing endonucleases (intron-homing) [57]. Mini-inteins are protein splicing elements that can ligate the polypeptide produced by the gene they disrupt post-translationally [58]. Their main difference from large inteins is the absence of a homing endonuclease domain, which enables large inteins to catalyse their self-excision [59]. HNH homing endonuclease genes are common in phage genomes [60] and are sometimes expressed to exclude other competing phages [61]. In the genomes of all five phages some of the HNH homing endonucleases reside next to hypothetical proteins, which may hint at the existence of spliced genes. Group I introns and inteins have already been discovered in phages, with the closest examples those of *Staphylococcus*, *Bacillus* and *Lactococcus* phages, and cases of both splicing DNA polymerase genes have been described there [62,63]. Regarding *Lactobacillus* phages, JCL1032 of *Lactobacillus delbrueckii* is the only one proven to have group I introns so far [64]. On the other hand, many *Lactobacillus* phages have genes for HNH endonucleases, with phage P2 predicted to have as many as eight in its genome (accession number KY381600.1) and phage LP65 of *L. plantarum* predicted to have two (accession number NC_006565.1).

#### 3.2.4. Morphogenesis, DNA Packaging, and Membrane Transport

Based on in silico analyses, the proteins that form the virion of the five *Lactobacillus* phages encompass a portal protein, a major capsid protein, a tail sheath protein, three unidentified tail proteins, and two baseplate proteins. The small and large subunit of a phage terminase are key enzymes of the DNA translocation and head filling [65]. A terminase DNA packaging subunit, characterised as the large subunit terminase, has been used to define the start of the five genomes here. These assigned functions were experimentally investigated by LC-MS/MS analysis on phage Dionysus and four virion-associated proteins were detected (Table 3). Moreover, twelve more non-virion-associated proteins of Dionysus were verified (Appendix A) due to the sensitivity of the proteomic analysis. Regarding the virion-associated proteins, in silico predictions were corroborated for the major capsid protein (peg. 170) and the portal protein (peg. 2). The other two proteins were a hypothetical protein in the vicinity of a baseplate protein (peg. 151) and a hydrolase (peg. 5) that could potentially be a virion-associated peptidoglycan hydrolase. However, the hypothetical protein and the hydrolase had low protein false discovery rate confidence (FDR < 1%), thus the uncertainty of these protein discoveries is high.

#### 3.2.5. Cell Wall and Membrane Degradation

A phage lytic cycle ends with the burst of the host cell and the release of progeny phages. The burst is accomplished due to the action of two groups of enzymes, holins and (endo)lysins. Holins are responsible for the formation of transmembrane holes, which allow lysins to reach and degrade the cell wall [66]. Out of the five different classes of lysins, the *Lactobacillus* phages studied thus far utilise only two, muramidases and amidases [67]. Holins usually have two or three transmembrane domains, although a few cases of holins with just one transmembrane domain have also been found [68]. For *Lactobacillus* phages there are examples of two- or three-transmembrane domain holins [69,70]. A TMHMM search coupled with DELTA-BLAST attributed traits typical of a holin to a hypothetical protein in all five phages (protein accession numbers: AYH92097, AUV60238, AUV59966, AYH91927, AYH92270). This hypothetical protein is right upstream of the lysin gene and was predicted to have a pair of transmembrane domains and low similarity to a holin of *Pediococcus pentosaceus*. All five phage genomes in this study possess at least one lysin. At the same time, the absence of lysogeny-related genes (integrases, excisionases and repressors of phage reproduction and lysis genes) indicates that these phages are most likely purely virulent [71]. However, since less than 24% of the genes in the five phages could be ascribed a function, and since some genes encoding hypothetical proteins were located at the antisense strand, it is possible that lysogeny-related genes remain undetected. New infections are initiated by the recognition and irreversible attachment of progeny phages to host bacterial receptors. The genomes of Bromius, Dionysus and Iacchus feature a glycerophosphodiester phosphodiesterase that has been described as a baseplate component of various bacteriophages against gram-positive bacteria, including *L. delbrueckii* phages [72]. Notably, this enzyme seems to facilitate the attachment of phage *Ld17* of *L. delbrueckii* by degrading glycerophosphodiesters of the cell envelope [73]. Owing to glycerophosphodiester phosphodiesterases and other degrading proteins, receptor-binding proteins can access bacterial receptors masked by surface molecules and catalyse phage adsorption [74]. Peptidoglycan-degrading enzymes of the phage tail tip are indispensable for phage DNA ejection which occurs after adsorption [73]. We presume that an annotated tail protein occasionally predicted as putative peptidoglycan hydrolase, which lies near a tail protein in the genomes of the five phages (AYH91914 of Iacchus, AYH92085 of Dionysus, AUV59954 of Bacchae, AYH92258 of Bromius, AUV60225 of Semele), is participating in this process [75].

#### 3.2.6. Other Predicted Proteins

In the region close to the DNA metabolism, replication, recombination, and repair genes, three coding sequences have been assigned a function but their exact purpose in the phage genome is unclear. One of them encodes a protein homologous with a DNA-binding ferritin-like protein from *Lactococcus lactis*, which provides protection against oxidative damage in bacteria [76]. The aerobic metabolism of *L. plantarum* leads to the production of H_2_O_2_ that the bacterium detoxifies via a manganese catalase (pseudo-catalase) [77]. Free Fe can interfere with the manganese complexes that are used by *L. plantarum’s* catalase to lower reactive oxygen levels [78]. Thus, the DNA-binding ferritin-like protein could help *L. plantarum* tolerate the hydrogen peroxide by binding to and storing excess free Fe that would otherwise damage the cell [79]. The second coding sequence encodes a protein that has homology with a low temperature requirement C protein of *Listeria monocytogenes*, which may be involved in lipid metabolism [80]. The protein of the third coding sequence is homologous with a YopX protein from *Bacillus subtilis*, an uncharacterised protein of bacteriophages and prophages of gram-positive bacteria [81]. This latter is solely produced by phages Dionysus and Iacchus.

### 3.3. The Diversity of the Five Phages Supports the Introduction of a New Lactobacillus Phage Genus

According to BLASTn, all five phages of this study show a low degree of similarity to Lactobacillus phage LpeD (70% identity over 51–55% query cover) and an even lower similarity to *Lactobacillus phage LP65* (83–88% identity over 20–23% query cover). The criterion considered here as a first-hand indicator of high nucleotide similarity between phage genomes is the one recommended by ICTV. Namely, this criterion sets a threshold of >50% sequence similarity when multiplying the BLASTn query cover by identity [42]. Thereon, 24 phage records that returned some degree of similarity, as well as the five new phages of this study, were subjected to an all-against-all BLASTn comparison by Gegenees. In the resulting heatplot of phylogenomic data (Figure 2), the phages presented in this study fall into one group that is separate from the other 24 phages. The calculated score distances within this group, which are equal to the average normalised similarities of fragments by BLAStn, were 44.92% or higher at a nucleotide level.

The Gegenees distinction is consistent with the outcome of the major capsid protein and large subunit terminase trees (Figure 3A,B). In both phylogenetic trees phages Semele, Bacchae, Iacchus, Dionysus and Bromius clearly cluster together and in the same clade with phage LpeD. In order to clarify if LpeD pertains to the same genus as the five new phages, we have chosen to include it in the remaining comparisons. 

Total protein checks undertaken with CMG-biotools system revealed significant homology among the proteomes of the five phages, while the homology between each of the five phages and phage LpeD was consistently less than 30% (Figure 4). Furthermore, homology within proteomes of Semele, Iacchus, Dionysus and Bromius was traced to the family of transposases and one or two other hypothetical protein families using MAFFT and BLASTp (Appendix A). This is surprising, given that duplicates (paralogues) in the same genome are rather rare for double-stranded DNA phages [82]. Transposases are motors of genome plasticity and adaptation [83]. Phages of *L. plantarum* are parasites of a versatile bacterium that can grow and survive in a raft of environmental niches. Hence, the abundance of transposases may assist phages Semele, Iacchus and Dionysus, which have two paralogous transposases, to adapt to novel niches and hosts by facilitating horizontal gene transfer with their hosts upon infection [84,85]. Moreover, the over-representation of transposases in the genomes of all five phages could reflect an abundance of transposases in the genome of host bacteria [86]. This scenario is consistent with the nature of the organic household waste samples, where a plethora of nutrients could translate to higher microbial densities, and thus higher rates of DNA exchange, as reported for bacteria from the Baltic Sea [60]. We investigated whether the transposases of the five new phages have a bacterial version by reviewing the BLASTp results. It turned out that all examined transposases showed significant homology with more than 20 different *Lactobacillus* sp., but unexpectedly not with their immediate host, *L. plantarum*.

Finally, yet importantly, Easyfig comparisons revealed a striking conservation of gene order and nucleotide homology amongst the five new phages. When these were compared to phage LpeD, we could still observe gene order conservation to a large extent, but the nucleotide homology was generally low (top genome; Figure 5). Concerning LP65, the genome of this phage displayed relatively low synteny and nucleotide homology against the five phage genomes described here (bottom genome; Figure 5). Given the distinctive characteristics of these five phage isolates, we propose a new *Lactobacillus* phage genus, that we term “Semelevirus”. The genus “Semelevirus” has as members the phages Semele (founder), Bacchae, Iacchus, Dionysus and Bromius.

## 4. Outlook

*Lactobacillus* phages have been only moderately addressed in comparison to other LAB phages. However, the doubling of publicly available, complete-genome sequences from 2017–2018 could imply a boom in this field of research. The many advantages displayed by their hosts, and especially by *L. plantarum*, may justify the fueled interest in these phages. There are quite some arguments to support the continuation of this trend in the coming years. *Lactobacillus* phages, as with other notorious LAB phages, can preclude obtaining high quality fermentation products on a regular basis. Hence, a systematic analysis of existing and newly-discovered *Lactobacillus* phage genomes would guarantee a more thorough understanding of their origin, evolution, and relationships with other phages [8]. Eventually, better knowledge on the phages of *Lactobacillus* could help to choose starter strains, and adjunct or aroma cultures with different phage sensitivity for strain rotation, and develop more efficient phage cocktails for inhibition of unwanted bacterial growth. At the same time, we believe that improved classification schemes are crucial to aid phage applications or to design efficient control strategies against phages, for example, by selecting mutants insensitive to infections of a range of phages. Should we expect an increasing number of *Lactobacillus* phages in the future, this need is even more urgent.

Looking at *L. plantarum* phages, we can discern some cases where steady accumulation of knowledge is essential. As mentioned, *L. plantarum* is an emerging biocontrol agent, as well as biostimulation agent of crops. Even if phages can impede successful application of this bacterium in the field they can still be sufficiently anticipated and tackled when we know enough about their biological profile. The same is true for *L. plantarum* starters used in the food industry. On the other hand, *L. plantarum* phages could be valuable biocontrol tools for the food industry [87]. First, they could be used to prevent growth of their host in those biotechnological processes in which *L. plantarum* is undesirable, such as in meat, beer, wine, or orange juice [7]. For example, phage cocktail preparations could minimize the addition of sulfur dioxide in those types of alcoholic beverages that should not undergo MLF. Studies on phages of beer spoiler strains of *Lactobacillus brevis*, *Lactobacillus paraplantarum* and *Pediococcus damnosus* have already introduced the concept of adding phages to inhibit these bacteria [88,89], [90]. Secondly, phage cocktails could offer a means to manipulate the MLF in wine, cider, and sour beer. Winemakers would have the chance to experiment by adding phages in aliquots of the same initial product but at different stages of the MLF and later evaluating the effects in each of the final wine products. Such an intervention may give rise to wines with novel characteristics, since through bacterial lysis phages can induce the release of enzymes for flavour development [91,92].

## Figures and Tables

**Figure 1 viruses-11-00611-f001:**
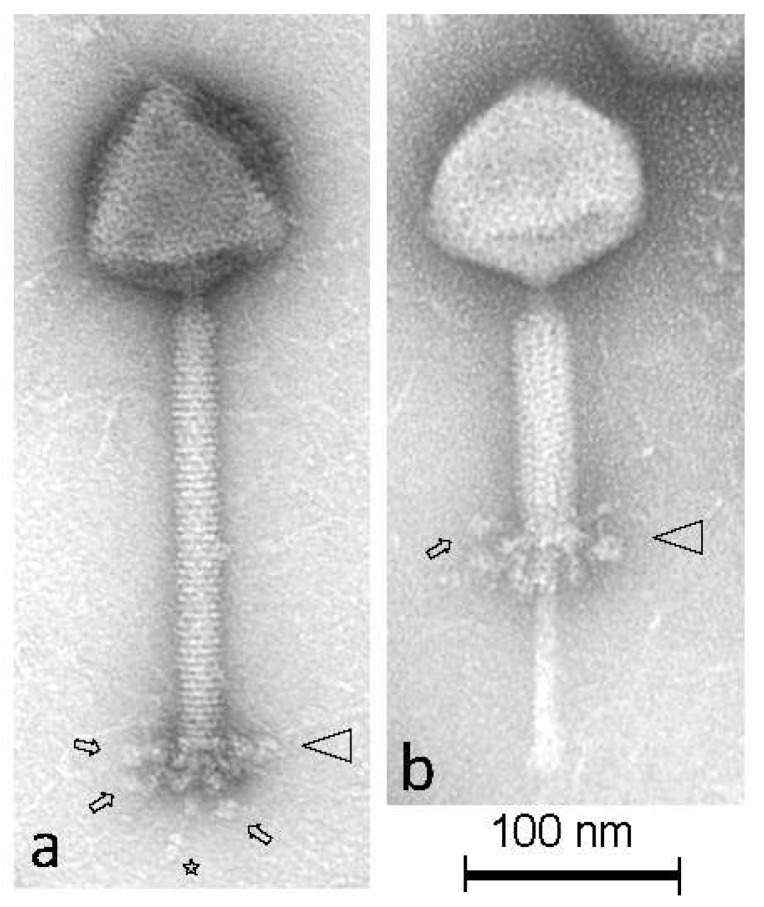
Transmission electron micrographs of *L. plantarum* phage Dionysus negatively stained with 2% (*w*/*v*) uranyl acetate. Triangles and arrows indicate the terminal baseplate structure and representative flexible appendages with terminal globular structures, respectively, attached underneath them. The single distal tail fibers terminating with three distinct knob-like structures are indicated by the star symbol. Phage Dionysus is shown with extended tail sheath (**a**) and contracted tail sheath (**b**), substantiating that it is a *Myoviridae* phage.

**Figure 2 viruses-11-00611-f002:**
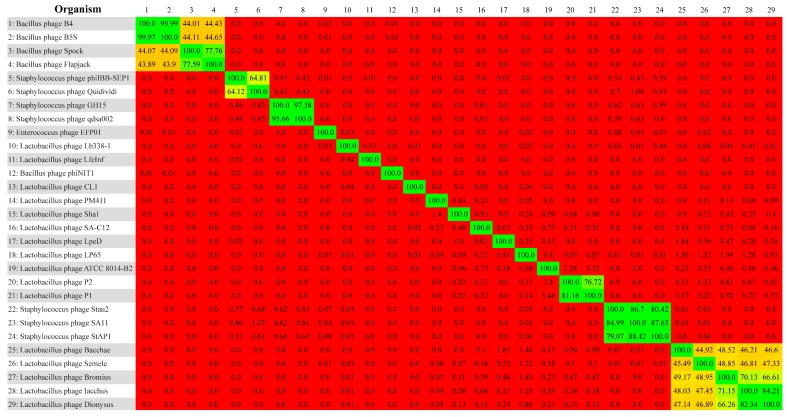
BLASTn heatplot generated using Gegenees. All-against-all comparisons were run with fragment length 50 bp, step size 25 bp, and threshold 0%. The studied phages (25–29) form a separate group.

**Figure 3 viruses-11-00611-f003:**
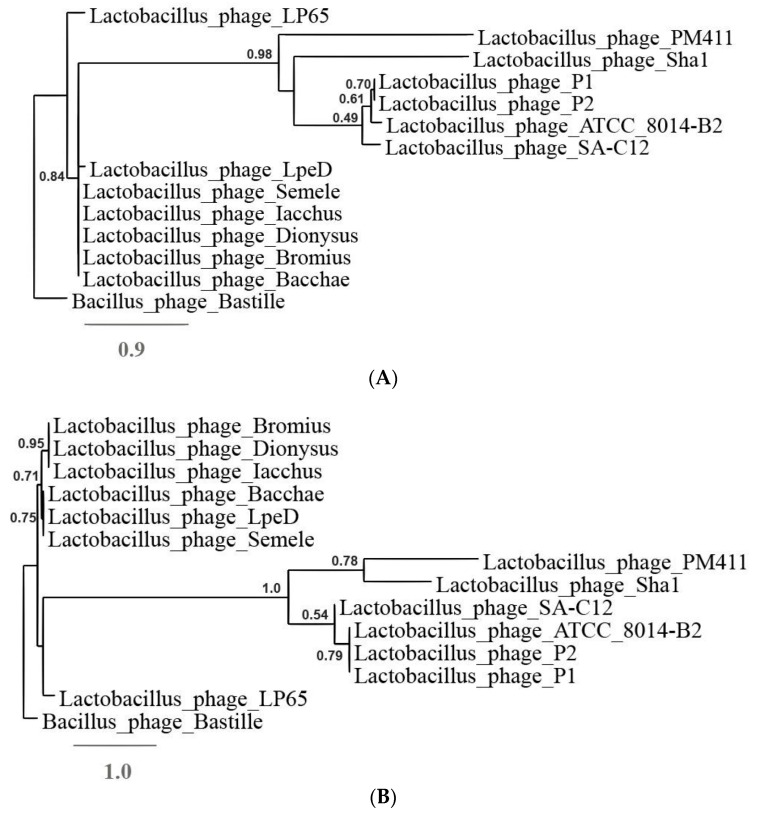
Phylogenetic trees for phages Semele, Bacchae, Dionysus, Iacchus and Bromius and other *Lactobacillus* phages yielding average similarity of at least 0.05 or higher with Gegenees. (**A**) Tree constructed using the major capsid proteins, (**B**) tree constructed using the large subunit terminases. The amino acid sequences were compared with the “One Click mode” (http://phylogeny.Lirmm.fr/). *Bacillus phage Bastille* proteins were used as outliers.

**Figure 4 viruses-11-00611-f004:**
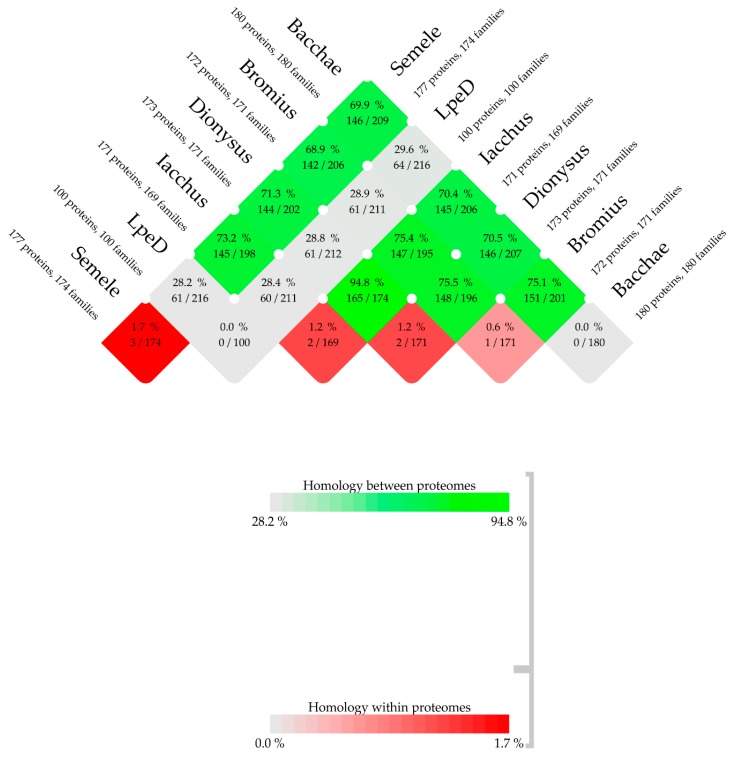
BLASTp comparison between and within proteomes of the five phages and their closest phage relative LpeD using CMG-biotools system. Phage pairs with >50% proteome homology are shown in green and those with <50% are depicted in grey. Red signifies the presence of one or more groups of paralogous proteins within a phage proteome.

**Figure 5 viruses-11-00611-f005:**
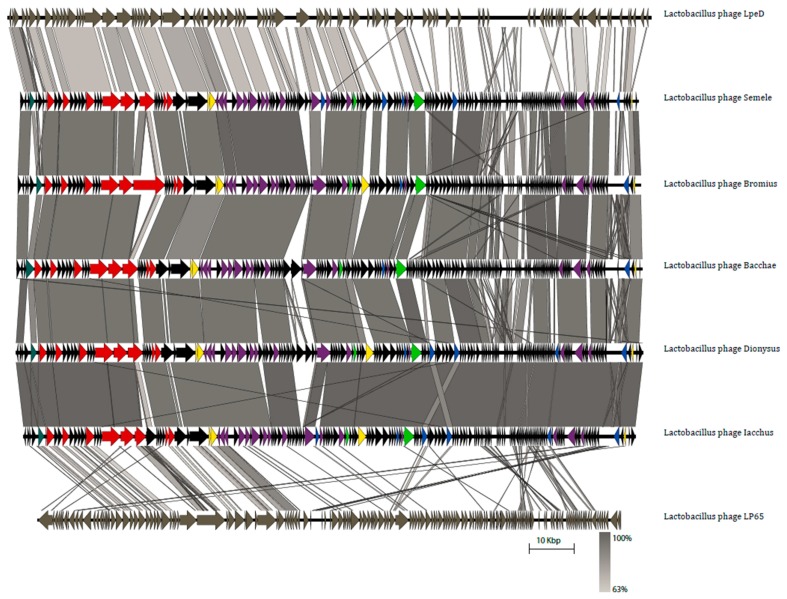
BLASTn comparisons of the genomes of the five phages with distant relatives LpeD and LP65 using Easyfig. Arrows represent the locations of genes and lines represent the level of homology between each tandemly placed group of phages. In the genomes of the five phages, genes are coloured according to the assigned function of the protein each gene encodes. These are: turquoise for DNA packaging, yellow for cell wall and membrane degradation, green for transcription and translation takeover, and blue for selfish genetic elements. Genes in red are part of the morphogenesis module and genes in deep purple are part of the DNA metabolism, replication, recombination and repair module. Black colour denotes genes that encode either hypothetical proteins or other predicted proteins with unclear functions. Appendix A follow the same colour scheme.

**Table 1 viruses-11-00611-t001:** Dimensions of phage Dionysus as measured with TEM.

Structural Components	Dimensions (nm)	Counted Phage Particles
**Head Diameter**	92.7 ± 3.5	17
**Tail Length (with Baseplate)**	222.4 ± 9.0	17
**Tail Width**	20.9 ± 0.6	17
**Baseplate Width**	44.1 ± 3.9	16
**Baseplate Length**	25.0 ± 2.4	16
**Baseplate Appendages Length**	21.8 ± 1.8	10
**Globular Structure Diameter**	9.2 ± 1.3	12
**Tail Fiber Length**	30.4 ± 2.2	13

**Table 2 viruses-11-00611-t002:** Basic genomic characteristics of the five *Lactobacillus* phage isolates.

Phage Isolate	Open Reading Frames with Assigned Function	Genome Size (bp)	G/C Content (%)	tRNA Genes
Semele	40/177	139,450	36.3	11
Bacchae	38/180	141,124	36.3	15
Iacchus	44/170	137,973	36.5	7
Dionysus	42/172	141,344	36.6	7
Bromius	39/172	140,527	36.5	7

**Table 3 viruses-11-00611-t003:** Mass spectrometry data for virion-associated proteins of phage Dionysus.

Description	FDR	Coverage (%)	No of Peptides	Molecular Mass (kDa)	Gene
Major capsid protein	High	2	1	53.5	peg. 170
Putative portal protein	High	2	1	60.9	peg. 2
Hypothetical protein	Low	3	1	101.9	peg. 151
Hydrolase	Low	5	1	35.9	peg. 5

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
