# Peer review of "Expanding the Diversity of Myoviridae Phages Infecting Lactobacillus plantarum—A Novel Lineage of Lactobacillus Phages Comprising Five New Members"

_viruses, 2019, doi:10.3390/v11070611_

Reviewer 1 Report

I have reviewed the revised manuscript submitted by Dr. Ifigeneia Kyrkou et al. titled “Expanding the diversity of Myoviridae phages infecting Lactobacillus plantarum - a novel lineage of Lactobacillus phages comprising five new members,” (Manuscript ID: viruses-408428) and I have provided my comments in the box.

I think authors answered my questions clearly and the manuscript has improved.

Please provide a citaten in line 181 , "(reference) " .

Author Response

Comments of Reviewer # 1 and answers from authors

I have reviewed the revised manuscript submitted by Dr. Ifigeneia Kyrkou et al. titled “Expanding the diversity of Myoviridae phages infecting Lactobacillus plantarum - a novel lineage of Lactobacillus phages comprising five new members,” (Manuscript ID: viruses-408428) and I have provided my comments in the box.

I think authors answered my questions clearly and the manuscript has improved.

Please provide a citaten in line 181, "(reference)".

Thank you for helping us improve our manuscript. We have now provided a citation in line 181 (now line 182).

Reviewer 2 Report

In the current manuscript, Kyrkou et al. describe five new phage genomes and suggest that a new genus is needed for these new phages.  

Some of the inquiries from the previous review were correctly addressed

- It is now clear why Dionysus was chosen as a reference phage

-The nature of the organic samples was also clarified

-The sequence fold is already available in the annotated genome

 However, I still found that this manuscript requires major revisions before publication. From my point of view, all aspects described above should be addressed to publish this paper.

 1-    Concerning different plaque morphologies, the authors described that “Plaques showing different morphologies were formed by phages against L. plantarum other than the phages presented in this study but that we isolated at the same time with the phages described here. Thus, the comment on various noted plaque morphologies is indeed not applicable here.”

If this is the case, it is unclear how plaques were selected for further purification and isolation of described phages. What was the criteria used to isolate the phages?

 2-    2.6 Identification of structural proteins (methods), 3.1. Isolation of Phages and Basic Features and table 2

The authors clearly show an effort to present MS data from phage Dionysus. It was a bit surprising the lack of data from the MS analysis in which only two proteins were found (and one peptide from each!). Knowing that the capsid protein is usually one of the most abundant proteins, many peptides are usually recognized. Also, the lack of tail peptides found is also uncommon.

It is described in methods that (line 146-148) “ 100 μL of the cesium chloride-treated phage lysate were transferred to Amicon Ultra filter unit (MWCO 30k Da) and centrifuged at 14.000 x g for 20 min and further desalted four times with 450 μL water”. If the protocol was correctly written, I am afraid that most of the phage sample was lost since that kind of centricon can only be centrifuged at 7500 x g for 10 min in a fixed angle centrifuge.  Have you measured protein concentrations in the flow through of the tube? Also, no SDS gel was performed. Centrifugation at high speed for 30 min in an Eppendorf tube is usually sufficient to concentrate phage at the bottom of the tube and proceed with SDS-PAGE and mass spectrometry.

 3-    The authors suggest that “We have now clarified in the abstract that the transmission electron microscopy analysis was only conducted for one representative phage (Dionysus) and we have explained why we judge that one out of the five phages is sufficient. Specifically, we explained that we chose a representative phage because the results of the comparative genomics and in silico protein predictions showed that there is a high degree of similarity among the five phages, which extends to all predicted structural proteins.” Even they could be right, I suspect that some interesting differences might be found. For instance, regarding a position in the genome putative protein number AUV60237.1 from Semele and putative protein number AYH92260.1  from Bromius (as just two examples) could be assigned as tape measure proteins (even though gene assignment in your sequences are “unknown”, TMP are usually hard to be assigned). If this is the case, tail length could be completely different from each other and easily seen by TEM. I strongly suggest TEM analysis for all phages is needed, especially if you are characterizing a new genus.

 4-    Figure 5. Blast comparison should be carefully revised. It is surprising than even genome identity is very high, some genes are not assigned to the same module. For example, Protein ID"AYH92258.1" from Bromius was identified as putative peptidoglycan hydrolase while protein ID "AUV60225.1" from Semele was identified as tail protein. These two proteins are the same except for the N-terminal of Semele protein that it is a bit longer. It is quite clear that both are part of the tail structure and differences could appear because of wrong gene assignment (wrong start codon for instance). It is also not surprising that tails components have peptidoglycan hydrolase activity. I only highlighted one gene as an example, please consider a deep revision of all gene assignments.  

Author Response

Comments of Reviewer # 1 and answers from authors

 I have reviewed the revised manuscript submitted by Dr. Ifigeneia Kyrkou et al. titled “Expanding the diversity of Myoviridae phages infecting Lactobacillus plantarum - a novel lineage of Lactobacillus phages comprising five new members,” (Manuscript ID: viruses-408428) and I have provided my comments in the box.

 I think authors answered my questions clearly and the manuscript has improved.

Please provide a citaten in line 181, "(reference)".

 Thank you for helping us improve our manuscript. We have now provided a citation in line 181 (now line 182).

  Comments of Reviewer # 2 and answers from authors

 In the current manuscript, Kyrkou et al. describe five new phage genomes and suggest that a new genus is needed for these new phages.

Some of the inquiries from the previous review were correctly addressed

- It is now clear why Dionysus was chosen as a reference phage

-The nature of the organic samples was also clarified

-The sequence fold is already available in the annotated genome

However, I still found that this manuscript requires major revisions before publication. From my point of view, all aspects described above should be addressed to publish this paper.

 1- Concerning different plaque morphologies, the authors described that “Plaques showing different morphologies were formed by phages against L. plantarum other than the phages presented in this study but that we isolated at the same time with the phages described here. Thus, the comment on various noted plaque morphologies is indeed not applicable here.” If this is the case, it is unclear how plaques were selected for further purification and isolation of described phages. What was the criteria used to isolate the phages?

Thank you for explaining your point. We understand now why this part seems confusing and obscure for the reviewer. We will try our best to be as thorough as possible in our answer concerning the procedure followed.

Before the screening of all organic household waste samples (one type collected from the treatment plant A and two types from B) we treated them in three different ways. In summary these were: 1) plain filtering, 2) concentration with PEG and 3) overnight enrichment with the indicator strains and then filtering (please see § 2.2. for details on the methods used). So these resulted in: 3 (types of samples) x 3 (treatment procedures) = 9 different treatments which we considered as new different samples at the screening stage. Following that, the screening stage was done against the two indicator strains, L1 and MW-1. This meant that we finally had: 2 (strains) x 9 (different samples) = 18 different plates that we checked for plaques independendly. In each of the 18 different samples, we found only one of the phages reported in this manuscript along with other phages that formed other plaques of different morphologies (as we explained in our past response). Therefore, the selection criterion was indeed different plaque morphologies per agar plate as observed by bare eye.

One can argue that this method was quite conservative because in some cases we ended up isolating the same phage twice from different plates. However, that method was chosen in an attempt to capture, as much as possible, the diversity of the phage population within each organic household waste sample. It is also important to note here that the concentration of samples with PEG has proven unfruitful for the phages of this manuscript.

To support our answer to the reviewer, we would like to give specific information concerning the combination of treatment/-s that yielded each phage of the manuscript.

Bacchae: sample of plant A – plain filtering – strain L1

Bromius: sample type two of plant B – plain filtering – strain MW-1

Dionysus: sample type two of plant B – enriched with strains – strain MW-1 AND

 sample type one of plant B – enriched with strains – strain MW-1

Semele: sample type one of plant B – plain filtering – strain L1

Iacchus: sample type one of plant B – plain filtering – strain MW-1

We believe this level of detail would only burden the text and confuse the reader, so we avoided extended explanations about this matter in the text. Instead, we have improved the text with the sentence “Each one of the phages Semele, Bacchae, Iacchus, Dionysus and Bromius was isolated from a different in vitro treatment of the original organic household waste samples (treatment details in § 2.2.)” (lines 205-7).

Thank you for helping us make the procedure more transparent for the reader.

2- 2.6 Identification of structural proteins (methods), 3.1. Isolation of Phages and Basic Features and table 2.

The authors clearly show an effort to present MS data from phage Dionysus. It was a bit surprising the lack of data from the MS analysis in which only two proteins were found (and one peptide from each!). Knowing that the capsid protein is usually one of the most abundant proteins, many peptides are usually recognized. Also, the lack of tail peptides found is also uncommon. It is described in methods that (line 146-148) “ 100 μL of the cesium chloride treated phage lysate were transferred to Amicon Ultra filter unit (MWCO 30k Da) and centrifuged at 14.000 x g for 20 min and further desalted four times with 450 μL water”. If the protocol was correctly written, I am afraid that most of the phage sample was lost since that kind of centricon can only be centrifuged at 7500 x g for 10 min in a fixed angle centrifuge. Have you measured protein concentrations in the flow through of the tube? Also, no SDS gel was performed. Centrifugation at high speed for 30 min in an Eppendorf tube is usually sufficient to concentrate phage at the bottom of the tube and proceed with SDS-PAGE and mass spectrometry.

 We highly appreciate the reviewer’s effort to help us pinpoint the reason behind the lack of further data from the MS analysis. Following the reviewer’s point we have turned again to the user guide for these specific Amicon Ultra filter units to confirm our choice of relative centrifugal force (please find the guide here http://file.yizimg.com/340681/2011050916293032.pdf ). According to page 5 of this guide, 14.000 x g is the recommended centrifugal force for concentration spin, thus the reason behind lack of further data must lie elsewhere and not at the damage of centricons. Moreover, we have successfully used this same method to perform MS analysis for other phages that were handled at the same time as the phages of this manuscript (results of the rest of phages can be given should the reviewer wish to). This further supports that the centricons can handle well centrifugation at 14.000 x g for 20 min.

The reviewers also suggests to perform an SDS-PAGE analysis prior to MS analysis. This is a good recommendation for future improvement of our pipeline. On the other hand, our currently applied bottom-up proteomics UHPLC-MS methodology is able to disentangle a complex mixture of proteins as detailed in Section 2.6 (manuscript lines 145-175). At the UHPLC-MS platform, each sample batch is accompanied with complex quality control samples and in this case, it was a tryptic digest standard of a human cell-line (HeLa S3) which contains >15,000 proteins.

The propagation of the five phages of this manuscript has been a very demanding task. After trying several different propagation methods, we have finally managed to reach somewhat high titers (108-1010 PFUs/mL) using the methods mentioned in § 2.2. Following that, the CsCl purifications were in most cases yielding titers lower than what is accepted for processing with TEM analysis (<108 PFUs/mL titer needed) and definitely quite poor for MS analysis. Thus, the reason why we initially limited this analysis to TEM and phage Dionysus as a representative of the five phages was logistic. The fact that we managed to have some results from the MS analysis (that we performed following the reviewers request) must mainly be attributed to the sensitivity of the instrument used. We will continue our explanation in the next comment of the reviewer.

We would like to note here that, should the reviewer judge it essential we would appreciate further suggestions on how to better report the abovementioned negative results in the manuscript.

 3- The authors suggest that “We have now clarified in the abstract that the transmission electron microscopy analysis was only conducted for one representative phage (Dionysus) and we have explained why we judge that one out of the five phages is sufficient. Specifically, we explained that we chose a representative phage because the results of the comparative genomics and in silico protein predictions showed that there is a high degree of similarity among the five phages, which extends to all predicted structural proteins.” Even they could be right, I suspect that some interesting differences might be found. For instance, regarding a position in the genome putative protein number AUV60237.1 from Semele and putative protein number AYH92260.1 from Bromius (as just two examples) could be assigned as tape measure proteins (even though gene assignment in your sequences are “unknown”, TMP are usually hard to be assigned). If this is the case, tail length could be completely different from each other and easily seen by TEM. I strongly suggest TEM analysis for all phages is needed, especially if you are characterizing a new genus.

 We strongly agree with the reviewer on the point that some interesting morphological diversities among the five phages could be discerned should all phages be analysed by TEM. Indeed we could draw some conclusions about the nature of the somewhat homologous tail proteins AUV60228 (Semele), AYH92260 (Bromius), AUV59956 (Bacchae), AYH92087 (Dionysus) and AYH91916 (Iacchus), as the reviewer suggests.

We have tried to perform TEM analysis for the remaining four phages. Spreadsheet S1 of the Supplementary files gives virion dimensions for phages Semele and Bacchae for comparisons with Dionysus’s virion. Moreover, the following figure shows the results of this analysis and may give some proof of our earlier observations, i.e. that it is logistic reasons (phage instabilities and low titers even after several propagation methods were tested) that did not allow us to proceed with TEM analysis for all five phages.

Supporting figure: Transmission electron micrographs of L. plantarum phages Semele (a), Bacchae (b), Iacchus (c) and Bromius (d) negatively stained with 2% (w/v) uranyl acetate. Phages Semele (a) and Bacchae (b) are shown both with extended and with contracted tail sheaths. Phage Bacchae (b) particles were only found attached to undefined matrix material putatively originating from the (MRS broth) lysate. Phage Iacchus could only be detected in few clusters of phage particles and (empty) phage heads. In lysates of phage Bromius (d), only intact or broken phage capsids (of a head diameter: 88.7 +/- 1.6 (n=2)) were detected, indicating that the number of remaining infective phage particles was below the limit of detection for electron microscopy.

Fortunately, based on the new measurements of Semele’s and Bacchae’s virions that we enclose in the Spreadsheet S1, phage Dionysus appears to represent sufficiently the morphology of all five phages. This is true also because the vast majority of predicted virion-associated proteins are shown to be highly homologous with only exception the tail proteins mentioned above (see Fig. 5). Of course, if the above controversial tail proteins are the tape measure proteins then we would expect Bromius and to a lesser extent Iacchus to have tail lengths different than the other phages. However, other than the gene’s size, deep bioinformatics analyses and revisions of related literature gave no indication that these are tape measure proteins.

To sum up, we agree with the reviewer that TEM for all five phages would be ideal because it could unravel some interesting details, mainly regarding a potential difference in tail length for phages Bromius and Iacchus but we find ourselves unable to do so due to logistic problems with phage propagation. However, we believe that Dionysus can still generally represent well the morphology of all five phages as shown by Spreadsheet S1, now included in the submission. We believe that we can still contribute to the phage community by publishing these results with the right clarification. To this end, we have: 1) provided the Spreadsheet S1 for virion comparisons with two more phages of the group, and 2) improved the text of the manuscript. Below is a sample of our corrections:

Transmission electron microscopy imaging analysis was done with a representative phage, i.e. phage Dionysus (Figure 1, Table 1). A representative phage was chosen due to the five phages being quite unstable and difficult to propagate at high titers. Nevertheless, Dionysus can sufficiently represent the studied phages because the vast majority of morphogenesis genes of the five phages were proven to be highly homologous (please read § 3.3. and Spreadsheet S1 for comparisons with virions of phages Semele and Bacchae). The only obvious exception was one gene predicted to encode a tail protein in all five phages (AUV60228 of Semele, AYH92260 of Bromius, AUV59956 of Bacchae, AYH92087 of Dionysus and AYH91916 of Iacchus).” (lines 215-21).

4- Figure 5. Blast comparison should be carefully revised. It is surprising than even genome identity is very high, some genes are not assigned to the same module. For example, Protein ID" AYH92258.1" from Bromius was identified as putative peptidoglycan hydrolase while protein ID "AUV60225.1" from Semele was identified as tail protein. These two proteins are the same except for the N-terminal of Semele protein that it is a bit longer. It is quite clear that both are part of the tail structure and differences could appear because of wrong gene assignment (wrong start codon for instance). It is also not surprising that tails components have peptidoglycan hydrolase activity. I only highlighted one gene as an example, please consider a deep revision of all gene assignments.

Thank you for this comment. We had unfortunately followed a very rigid approach for protein predictions earlier. This approach was mainly based on agreement of protein predictions across at least two databases for each phage. We had overlooked results on phage synteny, which would have helped visualise potential homologues better. We have now performed an in-depth revision of gene assignments by involving, amongst others, vertical one-by-one checks across gene homologies of the five phages. The following table collects the most important corrections we made per genome.

Phage

Protein

Prior Version

Corrected Version

Semele

AUV60226

put peptidoglycan   hydrolase

tail protein

Semele

AUV60233

hypothetical protein

baseplate protein

Semele

AUV60081

hypothetical protein

putative hydrolase

Semele

AUV60112

hypothetical protein

HNH-homing endonuclease

Semele

AUV60190

hypothetical protein

nicotinamide   mononucleotide transporter

Bromius

AYH92241

hypothetical protein

hydrolase

Bromius

AYH92258

put peptidoglycan   hydrolase

tail protein with   corrected start

Bromius

AYH92260

hypothetical protein

tail protein

Bromius

AYH92291

hypothetical protein

putative hydrolase

Bromius

AYH92319

hypothetical protein

HNH-homing endonuclease

Bacchae

AUV59955

hypothetical protein

tail protein

Bacchae

AUV59956

hypothetical protein

tail protein

Dionysus

AYH92085

put peptidoglycan   hydrolase

tail protein

Dionysus

AYH92087

hypothetical protein

tail protein

Dionysus

AYH92118

hypothetical protein

putative hydrolase

Dionysus

AYH92152

phosphatase/phosphodiesterase

RNA ligase

Dionysus

AYH92149

hypothetical protein

HNH-homing endonuclease

Iacchus

AYH91898

hypothetical protein

hydrolase

Iacchus

AYH91914

put peptidoglycan   hydrolase

tail protein

Iacchus

AYH91916

structural protein

tail protein

Iacchus

AYH91947

hypothetical protein

putative hydrolase

Iacchus

AYH91977

hypothetical protein

HNH-homing endonuclease

Iacchus

AYH91979

hypothetical protein

nucleoside   deoxyribosyltransferase

Please note that, we have decided to annotate the controversial proteins AUV60228 (Semele), AYH92260 (Bromius), AUV59956 (Bacchae), AYH92087 (Dionysus) and AYH91916 (Iacchus) mentioned in the reviewers third comment as “tail protein”. This is because as we explained earlier, we only had the gene size as an indication that these proteins may be “tape measure proteins”.

We submitted our annotation corrections to GenBank on 04/06/2019, and we hope that they will be available soon. We have also corrected the text to agree with the improved annotations (see corrections in Table 2 and lines 28, 252, 270, 298, 329-30, 340-1, 343, 349, 370, 373, 384-6) and have modified Figures 5, S1, S2, S3, S4 and S5 accordingly.

This manuscript is a resubmission of an earlier submission. The following is a list of the peer review reports and author responses from that submission.

Round  1

Reviewer 1 Report

In the current manuscript, the authors describe five new phages that are related to each other but show a low degree of similarity to Lactobacillus LpeD and LP65. These new genomes broaden the spectrum of the Lactobacillus Myoviridae family. The authors suggest that a new genus is needed for these new phages.  Nucleotide identity, predicted proteasome, genome characteristics and phylogenetic analysis information support their proposal. However, analysis of phage genomics is not enough to publish in a peer-review journal. In vivo/ In vitro data needs to be included to explode the phage biology behind these novel phage sequences. 

Please, find below the major problems found in the manuscript

Major points

Point 1

Line 17-18 (Abstract) and Figure 2

-“We examined the morphology and diversity of five new L. plantarum  phages  via transmission electron microscopy, whole genome shotgun sequencing and in silico protein predictions”  

Authors claim to show the morphology of 5 new phages isolated from organic waste treatment plants but at the end only one phage (Dionysus) is shown in the result section. From my point of view, not finding the information that you previously summarized (abstract) is a serious flaw.

-Also, pictures a-d are all showing the same viral particles (Dionysus). One image with an extended tail sheath and one image with a contracted tail sheath is enough per phage.

-It is also not clear why Dionysus was the phage chosen as a reference since Semele was the founder.

Point 2

159-160 and Figure 1 

-“All five isolates formed clear plaques with a diameter of approximately 0,5 mm on their respective host strains L1 and MW-1 (Figure 1).

Figure 1 only shows plaques formed by phage Dionysus on MW-1. However, in line 95, authors describe that plaque morphology is different between phages. If that is true, data needs to be available.

-Figure 1 is not informative and publications don’t usually show plaque plates unless a peculiarity is shown. Also, some kind of reflection is seen in it. 

Point 3

-In silico information is shown concerning putative structural proteins but no in vitro results are described by the authors. A protein mass spectrometry analysis of viral proteins of all phages will be necessary in order to improve the manuscript. 

From my perspective, if the manuscript describes 5 new isolated phages, all of them should be described in silico and in vitro. Overall, the abstract does not reflect what it is found later in the manuscript. 

Additional inquires: What is the host range of these phages? Are all of them Ca2+ dependent? What is the burst size? Have the authors tried to isolate natural resistant strains and evaluate if other phages were able to infect the resistant cells? Are there any differences in the genomes that support these ideas?  This information will be valuable for future biotechnological approaches and will enrich the manuscript discussion. 

Minor points

-Please, do not use first-person pronouns when refer to isolated phages (lines 24, 240, 245, 246, 252, 259, 321, 325, 330, 334, 338, 348, 351, 352, 357, 360,363, 364)

-Line 115-125 No information was provided concerning fold coverage of each genome sequence (not found in Genbank either).

-Figure 6. Genes divided into modules will facilitate reading and future discussion.

-In order to use these phages against bacteria spoilers or as biological tools to improve organoleptic properties, product (phage) safety and a more detailed description from where these phages were isolated is needed

Reviewer 2 Report

     In the present study, Kyrkou et al. have presented a whole genome comparison of five phages of Lactobacillus plantarum that were isolated from organic waste treatment plants. The authors claim that these five phages belong to a novel genus, which they have named “Semelevirus.”

    Although the number of bacteriophages (number of particles) is reportedly 10-fold higher than the number of bacteria, the knowledge regarding the location where these phages exist, their morphology, and their genome structure is very poor. Therefore, isolating these novel phages from the environment and determining their morphology and genome structure will be very important functions in the field of phage research.

    Lactic acid bacteria are valuable bacteria for agriculture and food production, as the authors have mentioned; however, the objective of this research on the phages that kill lactic acid bacteria is unclear. The authors must state the importance of their research or state their rationale for believing that studying phages that kill these valuable bacteria is an important issue in the introduction as well as the abstract.

     I believe this manuscript requires major revisions before publication, as indicated in the attached MS word file.
